# *Lacticaseibacillus rhamnosus* Fermentation Ameliorates Physicochemical Properties, Physiological Activity, and Volatile and Non-Volatile Compounds of Mango Juice: Preliminary Results at Laboratory Scale

**DOI:** 10.3390/foods14040609

**Published:** 2025-02-12

**Authors:** Jinlin Fan, Weiling Guo, Zheng Xiao, Jiacong Deng, Feifei Shi

**Affiliations:** 1College of Food and Bioengineering, Fujian Polytechnic Normal University, Fuqing 350300, China; m15880180003@163.com (J.F.); dengjc810508@163.com (J.D.); 2Institute of Food Science and Technology, College of Biological Science and Technology, Fuzhou University, Fuzhou 350108, China; weilingguo2021@163.com; 3Institute of Food Science and Technology, Fujian Academy of Agricultural Sciences, Fuzhou 350003, China; ethyxwat@163.com; 4College of Food Science, Fujian Agriculture and Forestry University, Fuzhou 350002, China

**Keywords:** fruit juice, *Lactobacillus*, physicochemical properties, antioxidant, metabolomics

## Abstract

*Lacticaseibacillus rhamnosus* is a strain predominantly used for juice production because of its excellent fermentation characteristics and strong acid production capacity. However, the influence of *L. rhamnosus* on the quality of mango juice has not yet been determined. Therefore, the effects of *L. rhamnosus* FJG1530 on the physicochemical properties, physiological activity, and volatile and non-volatile compounds of mango juice were extensively examined in this study. The data showed that *L. rhamnosus* FJG1530 possessed strong adaptability to mango juice, reducing its total sugar and increasing its total flavonoids. *L. rhamnosus* FJG1530 fermentation enhanced the ability of mango juice to clear the free radicals ABTS and DPPH, as well as improving the inhibition of lipase and α-glucosidase. In addition, *L. rhamnosus* FJG1530 treatment improved the volatile compounds in mango juice, especially promoting the formation of acids and alcohols. Simultaneously, metabolomic analysis revealed that 592 non-volatile compounds in mango juice were significantly changed by *L. rhamnosus* FJG1530 fermentation, with 413 dramatically increased and 179 significantly decreased metabolites. This study demonstrates that the fermentation process using *L. rhamnosus* FJG1530 was beneficial for ameliorating the quality of mango juice.

## 1. Introduction

Mango, also named *Mangifera indica*, is an important fruit in the Irvingiaceae family. It is commonly distributed in China, the Himalayas, India, Bangladesh, Mainland Southeast Asia, and the Malay Peninsula. According to an investigation by the Food and Agriculture Organization, in 2021, the planting area of mangoes in China reached 38.6 thousand hectares, with a yield of approximately 3.9 million tons [1]. Mango is a highly popular tropical fruit in the world because it possesses an attractive aroma, a bright color, a delicious taste, and a high nutritional value. Some studies confirmed that active substances derived from mangoes can prevent and treat many chronic diseases, including colitis, obesity, and liver injury [2,3]. Due to its high abundance of nutrients and its short harvest time, mango juice can serve as an appropriate medium for propagating microorganisms [4], which reduces its shelf life. Therefore, mango is universally processed into products, including juice, puree, slices in syrup, nectar, leather, pickles, canned slices, and chutney [5]. Among them, mango juice is becoming increasingly popular as a non-alcoholic beverage that allows consumers to enjoy the taste of mango all year round, just like eating raw mango. However, the flavor and taste of mango juice are relatively monotonous, resulting in a limited variety of products and a low bioavailability of nutrients.

Fermentation technology is regarded as an effective and environmentally friendly processing method, and fermented foods have been welcomed around the world. Fermentation extends the shelf life of food and ameliorates the flavor of raw food materials by secreting some antibacterial substances and converting complex compounds into bioactive metabolites [6]. For example, *Wickerhamomyces anomalus* fermentation elevates the level of polyphenols and flavonoids in red raspberry juice and reduces the level of total sugars and reducing sugars [7]. The changes in the physicochemical composition and the volatile and non-volatile compounds of mango juice induced by microbial fermentation also suggest a modification of its sensory characteristics [8]. In addition, some studies have confirmed that the antioxidant capacity, α-glucosidase inhibitory activity, and lipid inhibition rate of juice were enhanced after fermentation [9,10]. These can be important elements that affect consumers’ purchase preferences and the consumption of fermented mango juice produced at an industrial or semi-industrial level [11]. Lactic acid bacteria (LAB) are most important for some fermented foods, such as kimchi and yogurt. Fruits and vegetables are rich in sugar, vitamins, antioxidants, and other nutrients that facilitate the growth and reproduction of LAB; this makes them ideal substrates for lactic acid bacteria fermentation. Currently, *Lactobacillus* and *Bifidobacterium*, especially *Lactobacillus*, are the most familiar LAB applied in fermented products. Various *Lactobacillus strains*, including *Lactobacillus plantarum*, *Lactobacillus casei*, *Lactobacillus acidophilus*, etc., have been used for the fermentation of fruits and vegetables. For example, *Lactobacillus* (*L. brevis* 182, *L. plantarum* 239, *L. paracasei* 502, and *L. fermentium* 252 from spontaneously fermented plant foods) fermentation improved the concentrations of total phenolic compounds, total flavonoids, and amino acids in citrus juice [12]. Recently, Liu et al. selected three *Lactobacillus* strains (*L. plantarum* NCU116, *L. acidophilus* NCU402, and *L. casei* NCU215 from pickled vegetables and human feces) and successfully fermented mango juice. This fermentation was characterized by an alteration in the juice’s physicochemical parameters, nutritional constituents, harmful substance concentration, and volatile compound effects [13], suggesting that mango juice is a suitable substrate for *Lactobacillus*. In our previous study, *Lacticaseibacillus rhamnosus* FJG1530 (*L. rhamnosus* FJG1530) was isolated from traditional Chinese fermented foods, but the influence of *L. rhamnosus* FJG1530 on the metabolites and nutritional value of mango juice was not reported.

The aim of the present study was to preliminarily explore the influence of *L. rhamnosus* FJG1530 fermentation on the nutrition, flavor, and active substances of mango juice. Furthermore, changes in the volatile compounds and non-volatile compounds in mango juice before and after *L. rhamnosus* FJG1530 fermentation were detected using untargeted metabolome analysis. The present research provides a novel method for the amelioration of the quality and physiological function of mango juice.

## 2. Materials and Methods

### 2.1. Materials and Reagents

Mangoes were purchased online from Guangxi Province, China. De Man, Rogosa and Sharpe (MRS) liquid medium and MRS broth medium were obtained from Haibo Co., Ltd. (Shanghai, China). ABTS free radical, DPPH free radical, pancreatic lipase, and α-glucosidase were provided by Yuanye Company Ltd. (Shanghai, China). Other chemicals and reagents were obtained from BioPike Biotechnology Co., Ltd. (Shanghai, China).

### 2.2. Strains and Cultivation

The *L. rhamnosus* FJG1530, stored in a freezer (Haier DW-86L51, Qingdao, China) at −50 °C, was revived at 25 °C and then inoculated into sterile MRS agar. It was cultured at 37 °C for 24 h. A single colony of *L. rhamnosus* FJG1530 was added to MRS liquid medium and cultured at 37 ± 1 °C for 24 h. This process was carried out twice. The fermentation broth was centrifuged (Eppendorf 5418 R, Hamburg, Germany) at 6000× *g* for 10 min at 4 °C. Finally, the *L. rhamnosus* FJG1530 cells were collected and cleaned in triplicate and then dissolved using sterile physiological saline.

### 2.3. Mango Juice Fermentation

Fresh mango was obtained, cleaned, pulped, and then smashed using high-speed blender (Midea MJ-PB40ST10, Foshan, Chian). The mango juice was collected through a 60-mesh sieve, pasteurized at 85 °C for 30 min, and cooled using an ice bath. Subsequently, 4.0 mL of *L. rhamnosus* FJG1530 was added to 96.0 mL of mango juice and transferred to incubator (37 °C, 48 h, Boxun GSP-9160MBE, Shanghai, China). This process was repeated three times. Samples were collected in different stages (0, 6, 12, 24 and 48 h) and stored in a −60 °C freezer (ThermoFisher 900, MA, USA). The control group consisted of mango juice without added *L. rhamnosus* FJG1530 fermentation.

### 2.4. Viable Cell Counts and pH Value

The cell counts of *L. rhamnosus* FJG1530 during the fermentation process (0, 6, 12, 24 and 48 h) were measured using the pour plate technique. In short, the sample was diluted to different concentrations with sterile physiological saline, and 1 mL of the diluted sample was transferred to MRS agar counting plates. The plates were incubated at 37 °C for 24 h. Plates with a total bacterial colony count ranging from 30 to 300 were counted. The pH value was measured using a pH meter (Titumum U9N model).

### 2.5. Physicochemical Properties

#### 2.5.1. Total Sugars Contents and Titratable Acidity Measurement

Total sugar concentration of the sample was measured by utilizing the phenol sulfuric acid method. Then, 0.005 mL of unfermented or fermented mango juice was thoroughly mixed with 0.495 mL of water and then added to a tube containing 2.5 mL of sulfuric acid. The sample was incubated at 37 °C for 10 min and cooled to approximately room temperature. The absorbance of the mixture at 490 nm was detected using a UV-vis spectrophotometer (Shimadzu, Tokyo, Japan). The titratable acidity of the sample was detected via titration with 1 M sodium hydroxide and presented as lactic acid equivalents per liter (g/L).

#### 2.5.2. Total Polyphenolics and Flavonoids Content Measurement

The total polyphenolic concentration was detected according to a previous study, with some modifications [14]. Specifically, 1.0 mL of the sample was mixed with 2.0 mL of 10% Folin-Ciocalteu reagent and incubated at 25 °C for 3 min. Subsequently, 2.0 mL sodium carbonate (15%, *w*/*v*) was added to the sample. The absorbance of mixture at 765 nm was measured using a UV-vis spectrophotometer (Shimadzu, Tokyo, Japan), and the results were displayed expressed as gallic acid equivalents (mg/L).

The total flavonoid level was detected using the aluminum chloride method, with minor modifications [10]. Briefly, 1.0 mL of sample was added to 5.0 mL of 50 mg/mL NaNO_2_ solution and reacted at 25 °C for 5 min. Subsequently, 1.5 mL of AlCl_3_ solution (10% *w*/*v*) was added and reacted at 25 °C for 5 min. After that, 2.0 mL of 2 M sodium hydroxide was added and mixed thoroughly. After 10 min, the absorbance of the mixture at 500 nm was detected using a UV-vis spectrophotometer (Shimadzu, Tokyo, Japan), and then the total flavonoid level was calculated and expressed as rutin equivalent (mg/L).

### 2.6. Physiological Activity

#### 2.6.1. ABTS Free Radical Clearance Rate

The ABTS free radical clearance rate of the mango juice was measured based on a previously reported method with minor modifications [15]. Specifically, 50.0 mL of ABTS solution (7 mM) and 50.0 mL of K_2_S_2_O_8_ solution (2.45 mM) were mixed. After allowing the mixture to stand in the dark for 16 h, an ethanol solution (80% *v*/*v*) was added to the reaction solution obtained in the previous step to adjust the absorbance of 0.70 at 734 nm. Then, 0.76 mL of the diluted ABTS solution was added to 0.04 mL of mango juice and mixed thoroughly. The mixture was allowed to react in the dark for 20 min. The absorbance of each sample at 734 nm was detected using a UV-vis spectrophotometer (Shimadzu, Tokyo, Japan). The ABTS free radical clearance rate was computed using Formula (1):(1)$$ABTS\;free\;radical\;clearance\;rate\left(\%\right)=\frac{A_{blank}-A_{sample}}{A_{blank}}\times 100$$

Among them, A_blank_ represents the absorbance of the reaction solution without the samples, and A_sample_ represents the absorbance of the reaction solution with the samples.

#### 2.6.2. DPPH Free Radical Clearance Rate

The DPPH free radical clearance rate of the mango juice was measured according to a previously reported method, with minor modifications [16]. Briefly, 0.1 mL mango juice was mixed with 0.1 mL DPPH (0.1 mM) and then stored at room temperature without light for 0.5 h. The absorbance of the sample at 714 nm was measured using a UV-vis spectrophotometer (Shimadzu, Tokyo, Japan), and the DPPH radical scavenging activities were calculated using Formula (2):(2)$$DPPH\;free\;radical\;clearance\;rate\left(\%\right)=\frac{A_{blank}-A_{sample}}{A_{blank}}\times 100$$

Among them, A_blank_ represents the value of the reaction solution without the test samples, and A_sample_ represents the value of the reaction solution with the test samples.

#### 2.6.3. Inhibition Rate of Lipase

The lipase inhibition rate of the sample was measured based on a previous reported method, with minor modifications [17]. In brief, 50 μL of sample was added to 50 μL of lipase solution and incubated at 37 °C for 30 min. Subsequently, 50 μL of NPC solution was added to the reaction solution and then incubated at 37 °C for 30 min. The absorbance was measured at 600 nm using a UV-vis spectrophotometer (Shimadzu, Tokyo, Japan), and the lipase inhibition rate was calculated.

#### 2.6.4. Inhibition Rate of α-Glucosidase

The α-glucosidase inhibition rate of the sample was detected based on a previous reported method, with minor modifications [18]. In brief, the sample, pNPG solution (4 mM), and α-glucosidase solution (1 U/mL) were mixed in a ratio of 5:10:2 and incubated at 37 °C. After 0.5 h of reaction, the absorbance value of each sample at 405 nm was detected using a UV-vis spectrophotometer (Shimadzu, Tokyo, Japan), and the α-glucosidase inhibition rate was computed.

### 2.7. Volatile Compounds Analysis

The volatile component of samples with or without *L. rhamnosus* FJG1530 fermentation was identified using HS-SPME-GC-MS based on a previously reported method [18,19]. In brief, 5 mL of unfermented/fermented mango juice was transferred to a headspace bottle containing 2 g sodium chloride and 1 μL 2-octanol (5 mg/L). The mixture was equilibrated at 50 °C for 0.5 h, and then the volatile components were extracted for 30 min using HS-SPME. The volatile components were measured using the GC-MS system (Thermo Electron Corporation, Waltham, MA, USA). The volatile compounds were detected by matching their mass spectra with those in the NIST11 and Wiley databases. Semi-quantification analysis was performed using the internal standard (2-octanol) according to Formula (3):(3)$$C\left(\mu g/mL\right)=A_{c}\times \frac{C_{is}}{A_{is}}$$

Among them, A_C_ is the peak area of the detected sample, C_is_ is the final content of 2-octanol in sample, and A_is_ is the peak area of 2-octanol in sample.

### 2.8. Non-Volatile Compounds Analysis

The supernatants of samples (0.2 mL) were collected through centrifugation (12,000× *g*, 15 min, Eppendorf 5418 R, Hamburg, Germany) and thoroughly mixed with 0.8 mL of 80% aqueous methanol. After 20 min of stillness, 0.8 mL of the resulting supernatant was collected and freeze-dried. The lyophilizate powder was dissolved in 0.2 mL of 80% acetonitrile solution (*v*/*v*) and centrifugated for 10 min at 14,000× *g* (4 °C). The supernatant was collected for the analysis of non-volatile components utilizing UHPLC-MS/MS based on a previously reported study [20]. For UHPLC-MS/MS analysis, mobile phase A was acetonitrile/water/formic acid = 95:5:0.001 (*v*/*v*/*v*), and mobile phase B was acetonitrile/isopropanol/water/formic acid = 47.5:47.5:5:0.001 (*v*/*v*/*v*/*v*). MS analysis was executed in both ESI+ and ESI- modes with a data-dependent MS/MS scanning method. Among them, the parameters were set according to a previous report [9]. The raw data (including peak alignment, peak identification, and deconvolution) from UHPLC-MS/MS were further treated with CD software 3.8 (Thermo Fisher Scientific Inc., Avenue Waltham, MA, USA).

### 2.9. Statistical Analyses

Each treatment was performed in triplicate, and the data are presented as the mean ± SD. Principal component analysis (PCA) and hierarchical clustering analysis were used to assess multivariate statistical analyses of volatile compounds from control and FJG1530. Additionally, PCA, partial least squares discrimination analysis (PLS-DA), orthogonal PLS-DA (OPLS-DA), sparse PLS-DA (SPLS-DA), volcano plot, and KEGG enrichment pathways were used to perform multivariate statistical analyses of non-volatile compounds from control and FJG1530. Statistical analysis was carried out in SPSS 25.0 (SPSS Inc., Armonk, NY, USA) for ANOVA and Duncan’s multiple-range test, with statistical significance set at *p* ≤ 0.05.

## 3. Results and Discussion

### 3.1. Alterations in Active Bacterial and pH Value

The *L. rhamnosus* strain is widely applied as a starter culture to fermenting fruits and vegetables due to its strong capacity to adapt to acidic media [21]. The cell count is a vital indicator for assessing the fermentation properties of LAB. As shown in Figure 1A, the initial colony count of *L. rhamnosus* FJG1530 in mango juice was approximately 7.21 lg CFU/mL, and the colony count of *L. rhamnosus* FJG1530 in the fermented mango juice gradually increased with the fermentation process. After 48 h of fermentation, the colony count of mango juice with *L. rhamnosus* FJG1530 fermentation reached the highest level, approximately 9.15 lg CFU/mL, suggesting that mango juice supplies a suitable medium for the reproduction of *L. rhamnosus* FJG1530. Meanwhile, the pH value of the sample gradually declined with the fermentation process (Figure 1B), which is strongly associated with the formation of organic acids (including acetic and lactic acids) during LAB fermentation, leading to a decrease in pH [22]. A lower pH in the juice can limit the growth of microorganisms, which is beneficial for protecting the quality of the juice [23]. Therefore, the reduced pH values are advantageous for elevating the shelf life of mango juice.

### 3.2. Alterations in Physicochemical Properties

Microbial metabolic activity is affected by the types and concentrations of nutrients in fruit juices [14]. Among these nutrients, sugar is particularly important as a carbon source for the growth and reproduction of LAB. As shown in Figure 2A, compared with raw juice, the total sugar content in mango juice with *L. rhamnosus* FJG1530 fermentation was significantly reduced from the initial 112.04 mg/mL (*p* < 0.01), which is consistent with the findings of Chen et al. (2024) [24]. Meanwhile, the titratable acidity of mango juice fermented by *L. rhamnosus* FJG1530 increased dramatically at the 48th h of fermentation (*p* < 0.01, Figure 2B). A previous study reported that *L. rhamnosus* possesses a strong capacity for acid production [25].

Mango is rich in polyphenolics and flavonoids, which possess a series of health benefits, including anti-inflammatory, antioxidant, anti-hyperlipemic, and immunomodulatory properties [26]. Consequently, the total polyphenolic and flavonoid levels in both the control and FJG1530 groups were measured. The concentration of total polyphenols in unfermented mango juice was lower than that in fermented mango juice, but no statistical difference was observed (*p* > 0.05, Figure 2C). It is possible that *L. rhamnosus* FJG1530 fermentation facilitates the separation of the insoluble-bound form, leading to an enhancement in the total polyphenolic level [27]. Furthermore, the total flavonoid concentration was dramatically increased in the FJG1530 group compared to the control group (*p* < 0.01, Figure 2D). The increase in the total flavonoid concentration is associated with microorganism-derived enzymes that degrade complex polyphenols [28]. These findings suggest that *L. rhamnosus* FJG1530 has a dramatic effect on the flavonoid content of mango juices.

### 3.3. L. rhamnosus FJG1530 Fermentation Improved the Physiological Activity of Mango Juice

#### 3.3.1. The Antioxidant Capacity of Mango Juice Was Enhanced by *L. rhamnosus* FJG1530 Fermentation

Free radicals are a major cause facilitating aging and the occurrence of diseases, including skin damage, liver dysfunction, colitis, and cancer [29]. The LAB fermentation process has an essential influence on the bioactivity changes in juice, especially the antioxidant activity. Due to the high cost of evaluating antioxidants in vivo, in vitro free radical scavenging experiments are commonly used for a preliminary assessment of the antioxidant capacity of substances. To assess the antioxidant ability of mango juices, the ABTS and DPPH radical clearance rates were determined. The ABTS and DPPH radical clearance rates in the FJG1530 group were dramatically enhanced compared with the control group (*p* < 0.01, Figure 3A,B). It was previously reported that LAB fermentation could elevate the antioxidant activities of barley juice, apple juice, and watermelon juice, which is in agreement with our results [30]. These data are largely attributed to the fact that LAB fermentation can enhance the utilization of polyphenol utilization and increase the levels of flavone compounds with proton donor features [31]. In general, the increases in total polyphenol and flavonoid content in mango juice fermented by *L. rhamnosus* FJG1530 likely explain the enhancements in the ABTS and DPPH radical clearance rates.

#### 3.3.2. The Glycolipid Metabolism Regulation of Mango Juice Was Enhanced by *L. rhamnosus* FJG1530 Fermentation

With the development of society and changes in diet, glycolipid metabolism disorder has become a major global public health challenge, which is strongly related to a range of chronic diseases, including hyperlipidemia, hyperglycemia, obesity, and type 2 diabetes. Pancreatic lipase, regarded as an essential digestive enzyme, plays an essential role in lipid digestion and absorption in the body. Under the action of pancreatic lipase, dietary fat can be converted into fatty acids and glycerol, which promotes the occurrence of hyperlipidemia [32]. Therefore, inhibiting the pancreatic lipase is an important method for preventing the digestion and absorption of lipids, which is beneficial for reducing the risk of glycolipid metabolism disorder. As shown in Figure 3C,D, *L. rhamnosus* FJG1530 treatment significantly suppressed the pancreatic lipase activity in mango juice and significantly elevated the inhibition rate of α-glucosidase (*p* < 0.01). The function of α-glucosidase is to hydrolyze terminal glycosidic linkages, and its inhibition is beneficial for preventing the absorption of monosaccharides [33]. The inhibition of α-glucosidase activity prevents the increases in blood glucose, which is beneficial for reducing the risk of diabetes and cardiovascular disease [34]. These findings indicate that mango juice with *L. rhamnosus* FJG1530 fermentation has the potential to improve glucose and lipid metabolism.

**Figure 3 foods-14-00609-f003:**
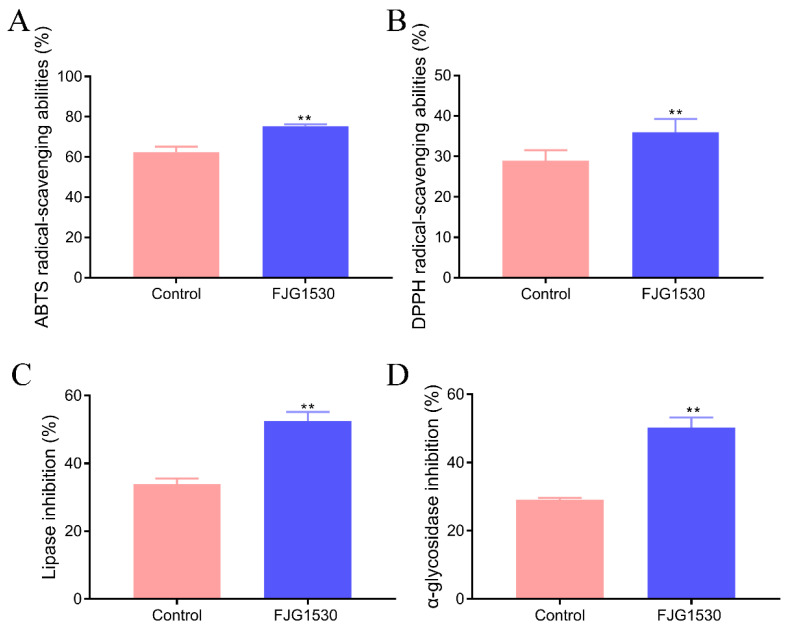
The changes in physiological activity of mango juice before and after *L. rhamnosus* FJG1530 fermentation. ABTS (**A**); DPPH (**B**); inhibition rate of lipase (**C**); inhibition rate of α-glucosidase (**D**). ** *p* < 0.01 compared with the control group.

### 3.4. L. rhamnosus FJG1530 Fermentation Influenced the Volatile Compounds in Mango Juice

Volatile compounds can influence the aroma and flavor of fermented juice, which affects consumers’ purchasing preferences. A previous study suggested that LAB fermentation has a notable influence on the formation of volatile compounds in fruit juice [35]. Therefore, the volatile components of mango juices before and after *L. rhamnosus* FJG1530 fermentation were measured. As shown in Figure 4, a total of 60 volatile compounds were measured across all samples, containing acids (19 kinds), alcohols (15 kinds), alkenes (9 kinds), esters (4 kinds), aldehydes (3 kinds), ketones (3 kinds), and others (7 kinds). Among them, the levels of acids in mango juice fermented with *L. rhamnosus* FJG1530 were elevated, which is consistent with the result of the titratable acidity. The formation of acids can suppress the reproduction of harmful bacteria, extending the shelf life of mango juices. In this study, the relative contents of hexanoic acid, 10,13-octadecadiynoic acid, tetradecanoic acid, 1,2,4-benzenetricarboxylic acid, n-hexadecanoic acid, 5-benzofuranacetic acid, benzoic acid, decanoic acid, n-decanoic acid, and nonanoic acid in the FJG1530 group were obviously increased compared with the control group. Among them, acetic acid possesses a sour and vinegary odor that is beneficial for the formation of a fermented juice flavor [36]. Furthermore, the relative contents of 3-cyclohexen-1-ol, cyclohexanol, 3-hexen-1-ol, 2-hexen-1-ol, and methanol in fermented mango juice were obviously increased compared to the control group. Alcohols are vital aromatic metabolites that widely possess unique floral and fruity odors [37]. Notably, the relative content of indole, vanillin, diphenyl ether and eicosanoic acid in the FJG1530 group was obviously lower than that in the control group.

To further observe the variation pattern of the volatile composition of mango juice before and after *L. rhamnosus* FJG1530 treatment, the alteration patterns of both unfermented and fermented mango juice were determined using the PCA score. As shown in Figure 5A, PC1 and PC2 displayed variances of 24.0% and 18.3%, respectively, totaling 42.3% of the total variance. The control group was mainly located in the second quadrant of the PCA score plot, while the FJG1530 group was mainly located in the third quadrant, suggesting that *L. rhamnosus* FJG1530 fermentation effectively altered the volatile composition of mango juice. A previous study also found that LBA fermentation could effectively change the overall composition of volatile compounds in other juices [38]. This PCA score plot result was corroborated by the result of hierarchical clustering analysis (Figure 5B). In detail, unfermented mango juice was related to 3-carene [F6], 3-hexen-1-ol [F12], benzaldehyde [F18], 2,6-octadien-1-ol [F29], diphenyl ether [F39], 1,2,3-propanetriol [F47], n-decanoic acid [F48], and 5-thiazoleethanol [F49], but fermented mango juice was related to propanoic acid [F3], 1-butanol [F5], (+)-4-carene [10], 1,5-cyclooctadiene [F11], 3-hexen-1-ol [F14], 3-cyclohexen-1-ol [F24], caryophyllene [F25], ethenone [F30], isoshyobunone [F34], ethyl maltol [F38], phenol [F42], sorbic acid [F44], 2(3H)-furanone [F45], benzoic acid [F50], methanol [F52], and decanoic acid [F56] (Appendix A). Among them, 3-Hexen-1-ol and 2-Hexen-1-ol were regarded as essential contributors to the volatiles of green and grassy notes in fruit juice [39].

### 3.5. L. rhamnosus FJG1530 Fermentation Also Changed the Non-Volatile Compounds in Mango Juice

Non-targeted metabolomics is widely applied to assess the alterations in non-volatile compounds between the control and FJG1530 groups. In this study, a total of 1659 non-volatile compounds in the mango juices were identified, including 1104 and 555 non-volatile compounds identified by ESI+ and ESI− modes, respectively. To analyze alterations in non-volatile compounds in the control and FJG1530 groups, PCA score analysis was performed. The first two principal components explained 64.7% and 93.8% of the total variances in the ESI+ and ESI- mode, respectively (Figure 6A and Figure 7A). The results of PCA score analysis suggest that the data from the same group clustered together, verifying the credibility and reproducibility of the test. In addition, the clear distinctions between the control and FJG1530 groups suggest the non-volatile compounds of mango juice were altered by *L. rhamnosus* FJG1530 fermentation. Some studies also found that LAB fermentation acts as an emerging food processing technology that helps to ameliorate juice quality [10,40]. Meanwhile, the results of PLS-DA (Figure 6B and Figure 7B), SPLS-DA (Figure 6C and Figure 7C), and OPLS-DA (Figure 6D and Figure 7D) were consistent with those of PCA.

To reveal the characteristics of non-volatile compounds between the control and FJG1530 groups, screening criteria, including a variable importance in projection (VIP) value of >1, a *p*-value of <0.05, and fold change > 2 or <0.5, were applied to select differential non-volatile compounds. In the ESI+ model, 273 non-volatile compounds in the FJG1530 group were dramatically altered compared with the control group, where 190 non-volatile compounds were increased and 83 non-volatile compounds were decreased (Appendix A). Among them, the relative contents of ilepcimide, (-)-nabilone, α-lactose, coumarin 106, lycocernuine, 2′_3′cyclicCMP, vihylpyrazine, choline, ensulizole, N6-Me-adenosine, D-glucosamine, eniluracil, eniluracil, 2′-deoxyadenosine, guanosine, phenacemide, 2′-deoxyinosine, and bacancosin were significantly decreased, but butabarbital, 2-furoylglycine, diethyl N-acetylglutamate, hexobarbital, epinephrine, Pro-Hyp, 3-succinoylpyridine, selva, 3-hydroxyquinine, perinaoprilat, betalamicacid, 2-methylhistamine, 3,5-dimethyl-4-benzylisoxazole, phenidone,1-methyladenine, 1,4-dihydroxy-2-naphthoate, DL-carnitine, 2-propylthiazolidine, 5′-deoxyadenosine, pyrazole, fenclonine, (R)-norcoclaurine, reducedrlboflavin, butoctamide, benserazide, N-α-L-acetyl-arginine, deanol, and DL-leucineamide were significantly increased. The 273 identified characteristic non-volatile compounds were matched to several metabolic pathways, including alanine, aspartate and glutamate metabolism, arginine metabolism, histidine metabolism, purine metabolism, tyrosine metabolism, lysine degradation, vitamin B6 metabolism, biotin metabolism, and lipoic acid metabolism (Figure 8A). In the ESI- model, a total of 319 non-volatile compounds in the FJG1530 group were significantly altered compared with the control group, including 223 increased and 96 decreased non-volatile compounds (Appendix A). Among these, 5-methoxytryptamine, L-2-methytryptophan, phenobarbital, acetyl-L-carnitine, oxagrelate, 5-methoxyindole, xanthine, zibotentan, ethyl-β-D-glucuronide, oxamate, uric acid, 1,5-DAN, cis,cis-muconic acid, palmitoleic acid, 1D-chiro-inositol, carbaryl, and 2_3-dimethylmaleate were significantly reduced, but 4-oxoproline, tioxacin, (+)-bornane-2_5-dione, 2′-deoxyadenosine, guanosine, N_N-dihydroxy-L-tyrosine, phaseolicacid, and vanillic acid 4-sulfate were significantly increased. The 319 identified characteristic non-volatile compounds were matched to several metabolic pathways, including alanine, aspartate and glutamate metabolism, glyoxylate and dicarboxylate metabolism, purine metabolism, nicotinate and nicotinamide metabolism, citrate cycle (TCA cycle), terpenoid backbone biosynthesis, glycine, serine and threonine metabolism, and starch and sucrose metabolism (Figure 8B). In summary, the above pathways were mainly associated with amino acid metabolism, which is consistent with a previous report [41]. Therefore, we speculated that amino acid metabolism is likely the major factor responsible for the alteration in non-volatile compounds in fermented mango juice.

## 4. Conclusions

The present study revealed the influence of *L. rhamnosus* FJG1530 fermentation on the physicochemical properties, physiological activity, and volatile and non-volatile compounds of mango juice. The results exhibited that *L. rhamnosus* FJG1530 fermentation drastically decreased the total sugar content in mango juice, while it drastically increased the total flavonoids. Meanwhile, the antioxidant, hypolipidemic, and hypoglycemic capacities of mango juice were enhanced by *L. rhamnosus* FJG1530 fermentation. In addition, *L. rhamnosus* FJG1530 fermentation improved the volatile compounds of mango juice, especially acids and alcohols. Untargeted metabolomics revealed that *L. rhamnosus* FJG1530 treatment drastically altered the metabolic profile, primarily related to amino acid metabolism. These data suggest that *L. rhamnosus* FJG1530 treatment is an effective strategy for improving the quality of mango juice. However, sensory analysis of mango juice with and without *L. rhamnosus* FJG1530 fermentation could not be conducted due to the limited samples size (100 mL). Further studies are necessary to explore the influences of *L. rhamnosus* FJG1530 on the sensory analysis of mango juice and to investigate consumers’ willingness to purchase and drink *L. rhamnosus* FJG1530 fermented mango juice, especially when produced at industrial or semi-industrial scales.

## Figures and Tables

**Figure 1 foods-14-00609-f001:**
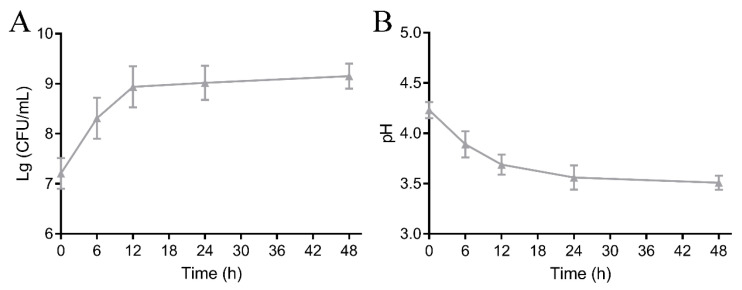
Alterations in viable counts of *L. rhamnosus* FJG1530 (**A**) and pH (**B**) of mango juice during the experimental process.

**Figure 2 foods-14-00609-f002:**
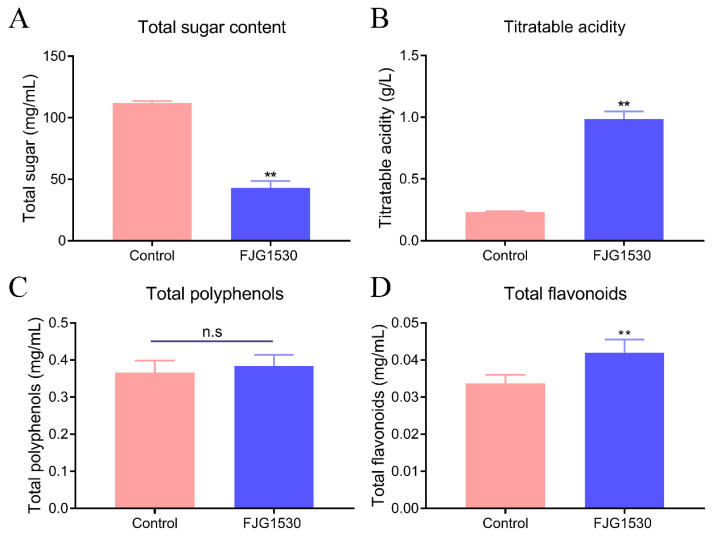
The changes in physicochemical properties of mango juice before and after *L. rhamnosus* FJG1530 fermentation. Total sugar (**A**), titratable acidity (**B**), total polyphenols (**C**) and total flavonoids (**D**). n.s means no statistical difference; ** *p* < 0.01 compared with the control group.

**Figure 4 foods-14-00609-f004:**
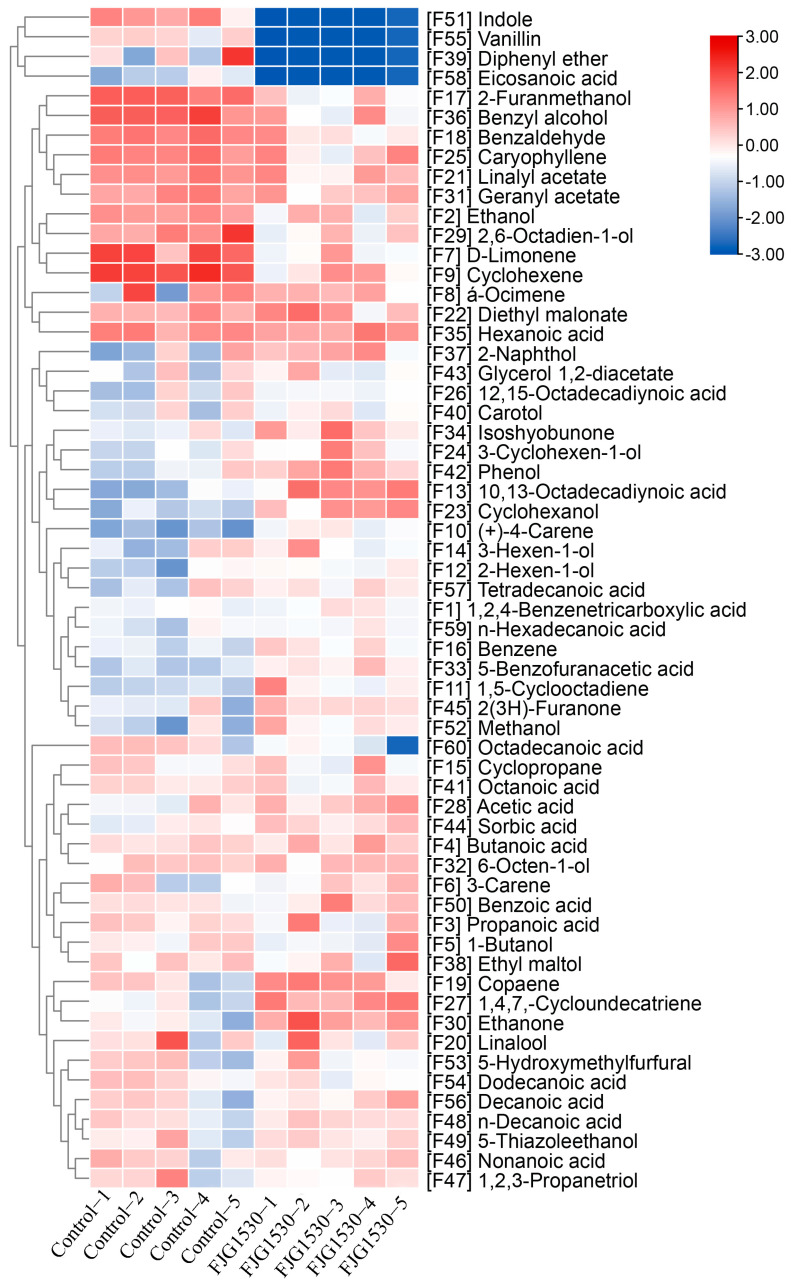
Change in volatile compounds in mango juice before and after *L. rhamnosus* FJG1530 fermentation.

**Figure 5 foods-14-00609-f005:**
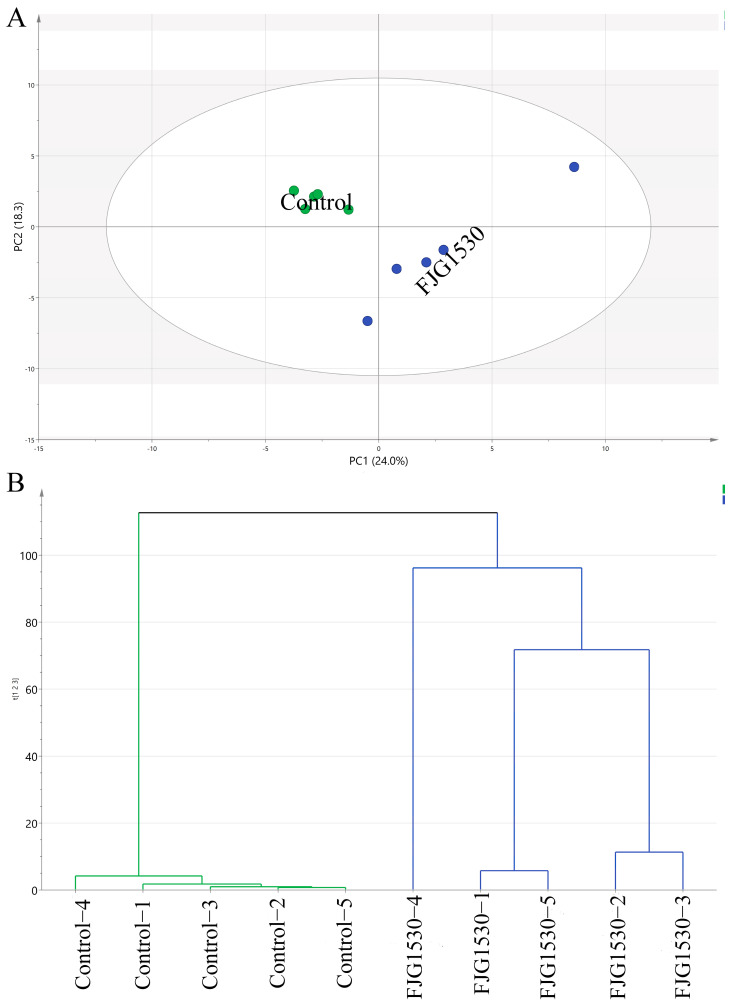
Principal component analysis (PCA) score (**A**) and hierarchical clustering analysis (**B**) applied to assess volatile compounds in the unfermented and fermented mango juice.

**Figure 6 foods-14-00609-f006:**
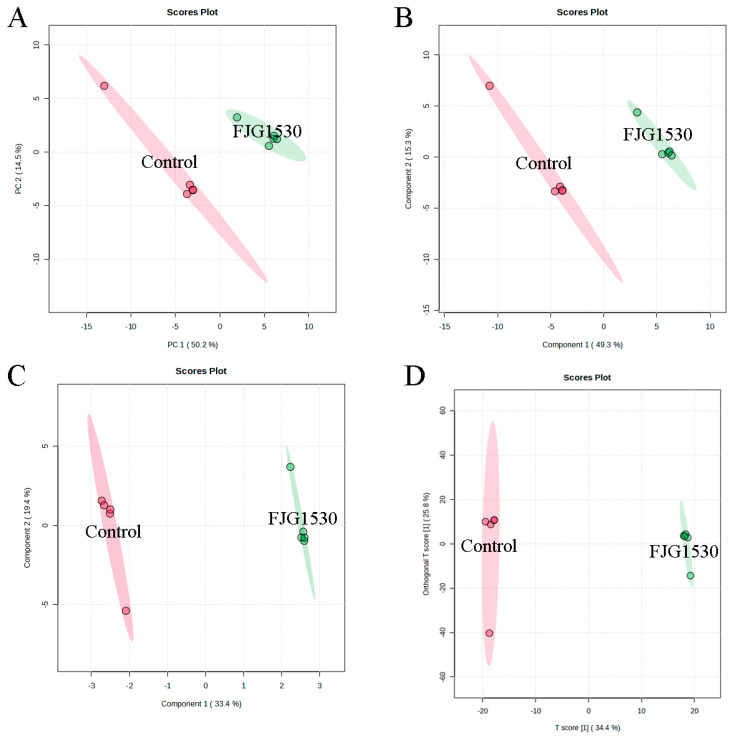
Alterations in non-volatile compounds of sample from the control and FJG1530 groups in the ESI+ model. PCA (**A**); PLS-DA (**B**); OPLS-DA (**C**); SPLS-DA (**D**).

**Figure 7 foods-14-00609-f007:**
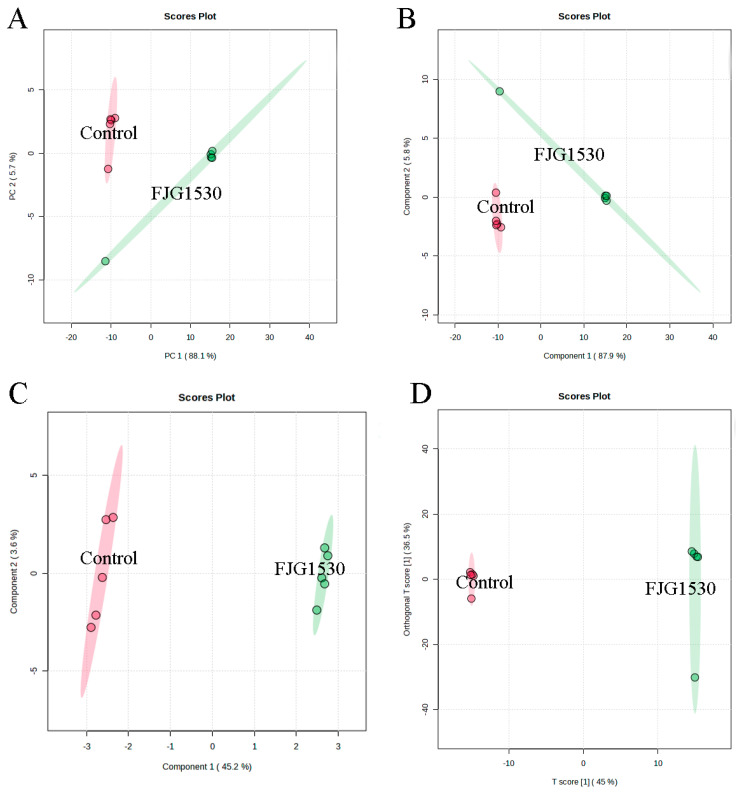
Alterations in non-volatile compounds of sample from the control and FJG1530 groups in the ESI- model. PCA (**A**); PLS-DA (**B**); OPLS-DA (**C**); SPLS-DA (**D**).

**Figure 8 foods-14-00609-f008:**
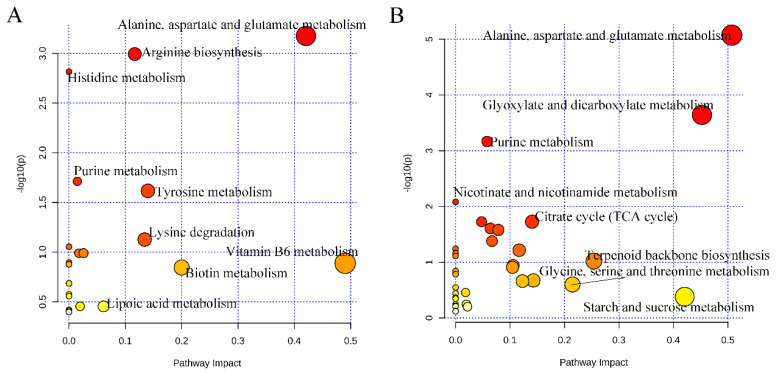
KEGG enrichment pathways of the differential non-volatile compounds in mango juice. The ESI+ model (**A**); the ESI- model (**B**). Colors (varying from yellow to red) mean the metabolites are in the data with different levels of significance.

## Data Availability

The original contributions presented in the study are included in the article/Appendix A. Further inquiries can be directed to the corresponding author.

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
