# Peer review of "Lacticaseibacillus rhamnosus Fermentation Ameliorates Physicochemical Properties, Physiological Activity, and Volatile and Non-Volatile Compounds of Mango Juice: Preliminary Results at Laboratory Scale"

_foods, 2025, doi:10.3390/foods14040609_

Round 1

Reviewer 1 Report

Comments and Suggestions for Authors

Dear Authors

In this study evaluated the effects of L. rhamnosus FJG1530 on the physicochemical properties, physiological activity, and volatile and non-volatile compounds of mango juice. However, I can send some observations to improve the manuscript:

Keywords

I recommend using different words than those that appear in the title.

Introduction

I recommend updating reference 5.

Materials and Methods

Pag. 3. Line 98. Freezer or refrigerator? Add name, model, and country of origin of equipment.

Pag. 3. Lines 99-102. I recommend using abbreviations appropriately. For example, replace "hours" to "h" and "minutes" to "min". Please review the entire document carefully.

Pag. 3. Line 101. Add name, model, and country of origin of centrifuge.

Pag. 3. Lines 107 and 112. Add name, model, and country of origin of high-speed blender and freezer.

Pag. 3. Line 110. Where did they place the flasks? Did they use an incubator?

Pag. 3. Line 118. Add name, model, and country of origin of incubator.

Pag. 3. Line 127. Add model and country of origin.

Pag. 3. Line 134. Use subscripts appropriately in chemical formulas.

Pag. 3. Line 135. Add name, model, and country of origin of UV-vis spectrophotometer.

Pag. 3. Line 142. Where was the absorbance measured?

Pag. 4. Line 145-165. Please add the equipment that was used to measure the absorbance.

Pag. 4. Line 168. I recommend updating reference 15, it is a widely used methodology.

Pag. 4. Line 172 and 178. Where was the absorbance measured?

Pag. 4. Line 174. Verify that all scientific names remain in italics.

Pag. 5. Lines 198 and 200. Add name, model, and country of origin of equipment.

Add the year of references 12, 18, 22, 35, 37 and 39.

Author Response

Comments to the Author

In this study evaluated the effects of L. rhamnosus FJG1530 on the physicochemical properties, physiological activity, and volatile and non-volatile compounds of mango juice. However, I can send some observations to improve the manuscript:

Comment 1:

Keywords

I recommend using different words than those that appear in the title.

Response:

We sincerely thank the reviewer for the positive and constructive comments. We have modified the key words (Fruit juice; Lactobacillus; Physicochemical properties; Antioxidant; Metabolomics) in the revised manuscript.

Comment 2:

Introduction

I recommend updating reference 5.

Response:

Thank you for your kind advice. We have revised reference 5 in the revised manuscript. (Line 482-484)

Comment 3:

Materials and Methods

Pag. 3. Line 98. Freezer or refrigerator? Add name, model, and country of origin of equipment.

Response:

Thank you for your kind advice. We have supplemented the detail inflammation of refrigerator in the revised manuscript (Line 99).

Comment 4:

Pag. 3. Lines 99-102. I recommend using abbreviations appropriately. For example, replace "hours" to "h" and "minutes" to "min". Please review the entire document carefully.

Response:

We are very sorry for the incorrect description in the manuscript. We have revised these descriptions in the revised manuscript. (Line 102,104, and 122).

Comment 5:

Pag. 3. Line 101. Add name, model, and country of origin of centrifuge.

Response:

Thank you for your kind advice. We have supplemented the detail inflammation of centrifuge in the revised manuscript (Line 103).

Comment 6:

Pag. 3. Lines 107 and 112. Add name, model, and country of origin of high-speed blender and freezer.

Response:

Thank you for your kind advice. We have supplemented the detail inflammation of high-speed blender and freezer in the revised manuscript (Line 109 and 114-115).

Comment 7:

Pag. 3. Line 110. Where did they place the flasks? Did they use an incubator?

Response:

We are very sorry for the incorrect description in the manuscript. We have revised this sentence in the revised manuscript. (Line 111-112)

Comment 8:

Pag. 3. Line 118. Add name, model, and country of origin of incubator?

Response:

Thank you for your kind advice. We have supplemented the detail inflammation of incubator in the revised manuscript. (Line 122)

Comment 9:

Pag. 3. Line 127. Add model and country of origin.

Response:

Thank you for your kind advice. We have supplemented the detail inflammation of UV-vis spectrophotometer in the revised manuscript. (Line 132)

Comment 10:

Pag. 3. Line 134. Use subscripts appropriately in chemical formulas.

Response:

Thank you for your kind advice. We have revised this description according to your comments. (Line 138)

Comment 11:

Pag. 3. Line 135. Add name, model, and country of origin of UV-vis spectrophotometer.

Response:

Thank you for your kind advice. We have supplemented the detail inflammation of UV-vis spectrophotometer in the revised manuscript. (Line 139)

Comment 12:

Pag. 3. Line 142. Where was the absorbance measured?

Response:

Thank you for your kind advice. We have supplemented the detail inflammation of measuring equipment in the revised manuscript. (Line 145-147)

Comment 13:

Pag. 4. Line 145-165. Please add the equipment that was used to measure the absorbance.

Response:

Thank you for your kind advice. We have supplemented the detail inflammation of measuring equipment in the revised manuscript. (Line 158-159 and 167-168)

Comment 14:

Pag. 4. Line 168. I recommend updating reference 15, it is a widely used methodology.

Response:

Thank you for your kind advice. We have updated reference 15 in the revised manuscript. (Line 510-511)

Comment 15:

Pag. 4. Line 172 and 178. Where was the absorbance measured?

Response:

Thank you for your kind advice. We have supplemented the detail inflammation of measuring equipment in the revised manuscript. (Line 177-178 and 184-186)

Comment 16:

Pag. 4. Line 174. Verify that all scientific names remain in italics.

Response:

Thank you for your kind advice. We have revised this sentence according to your comments. (Line 173, 181, 280, and 298)

Comment 17:

Pag. 5. Lines 198 and 200. Add name, model, and country of origin of equipment.

Response:

Thank you for your kind advice. We have supplemented the detail inflammation of centrifuge in the revised manuscript. (Line 205, and 208-209)

Comment 18:

Add the year of references 12, 18, 22, 35, 37 and 39.

Response:

Thank you for your kind advice. We have added the year of references 12, 18, 22, 35, 37 and 39 in the revised manuscript. (Line 505, 518, 529, 561, 565, and 570)

Reviewer 2 Report

Comments and Suggestions for Authors

The subject of the article is a manuscript entitled "Lacticaseibacillus rhamnosus Fermentation Ameliorates Physi-2 cochemical Properties, Physiological Activity, Volatile and 3 Non-Volatile Compounds of Mango Juice”.

The paper is well structured, methods and data, results and discussion are clearly described. The work certainly brings new knowledge to the subject area, which the authors have explained in the work.

I consider that the manuscript can be improved further. Thus, I recommend the revising of the article based on the main points mentioned below. Please replace in all manuscript ” hours” with ”h”; “minutes” with „min”. Please write in all manuscript in Italic font the name of L. rhamnosus (lines – 242, 262,280 etc.).

Line 98 - please replace “refrigerator” with  “freezer”.

Line 134 - please write the formula of Na2CO3 with subscript (Na2CO3)

Line 164 – please write with subscript all the terms from equations 1-3 (Asample, Ablank,)

Line 242 – please replace L. rhamnosu with L. rhamnosus

Figures 5, 6, 7 – please enlarge the axis labels of graphs.

Author Response

Comments to the Author

The subject of the article is a manuscript entitled "Lacticaseibacillus rhamnosus Fermentation Ameliorates Physi-2 cochemical Properties, Physiological Activity, Volatile and 3 Non-Volatile Compounds of Mango Juice”.The paper is well structured, methods and data, results and discussion are clearly described. The work certainly brings new knowledge to the subject area, which the authors have explained in the work.

Comment 1:

I consider that the manuscript can be improved further. Thus, I recommend the revising of the article based on the main points mentioned below. Please replace in all manuscript ” hours” with ”h”; “minutes” with „min”. Please write in all manuscript in Italic font the name of L. rhamnosus (lines – 242, 262,280 etc.).

Response:

Thank you for your kind advice. We have revised this sentence according to your comments. (Line 102, 122, 260, 280, and 298)

Comment 2:

Line 98 - please replace “refrigerator” with “freezer”.

Response:

Thank you for your kind advice. We have revised this sentence according to your comments. (Line 99)

Comment 3:

Line 134 - please write the formula of Na2CO3 with subscript (Na2CO3).

Response:

Thank you for your kind advice. We have revised this sentence according to your comments. (Line 138)

Comment 4:

Line 164 – please write with subscript all the terms from equations 1-3 (Asample, Ablank,)

Response:

Thank you for your kind advice. We have revised this sentence according to your comments. (Line 161-162, 170-171, and 202-203)

Comment 5:

Line 242 – please replace L. rhamnosu with L. rhamnosus.

Response:

Thank you for your kind advice. We have revised this sentence according to your comments. (Line 260)

Comment 6:

Figures 5, 6, 7 – please enlarge the axis labels of graphs.

Response:

Thank you for your kind advice. We have revised this sentence according to your comments. (Line 370, 392, and 434)

Reviewer 3 Report

Comments and Suggestions for Authors

Author Response

Comments to the Author

The paper “Lacticaseibacillus rhamnosus fermentation ameliorates physicochemical properties, physiological activity, volatile and non-volatile compounds of mango juice” seems to clearly illustrate in the title and in the abstract the subject of the study.The microorganism Lacticaseibacillus rhamnosus is a strain employed for juice production. It has a very good ability to improve the acidity of its fermented products, but its capacities have not yet been experimented with mango (Mangifera indica) juices. The authors employed the specific strain L. rhamnosus FJG1530 with a juice produced with mango cultivated in China.The results showed many important modifications in the chemical composition, the antioxidant abilities and the volatile profile of the fermented mango juices.The composition was analyzed employing headspace micro-extraction-solid- phase microextraction combination with gas chromatography mass spectrometry (HS- SPME-GC-MS) for volatile compounds and high performance liquid chromatography with quadrupole time-of- flight mass spectrometry (UPLC-Q-TOF-MS), for the non-volatile compounds. The references in the article are recent and relevant and there is not an excessive number of self-citations from the authors.

Comment 1:

The main weaknesses:

The authors carried out the essay only in a little quantity of mango juice (100 mL): they are preliminary tests at laboratory scale. However, they did not evidence this condition in the paper.

I suggest the authors change the title, to evidence that this is only a preliminary study, and it is necessary to emphasize it the abstract, in the aim of the work and in the conclusions that it is necessary to test the potentialities of the strain L. rhamnosus FJG1530 in a higher quantity of juice (maybe each juice essay should be 4-5 Liters) - and in a real industrial condition of mango juice production.

These are only suggestions for the title:

Lacticaseibacillus rhamnosus fermentation ameliorates physicochemical properties, physiological activity, volatile and non-volatile compounds of mango juice: preliminary results at laboratory scale.”

Or

“Preliminary results at laboratory scale of the positive effect of ” Lacticaseibacillus rhamnosus fermentation on the composition and physiological activity of mango juice.”

Response:

Thank you for your kind advice. We have modified the title of this manuscript according to your comment. (Line 2-5)

Comment 2:

The sensory aspects: the paper showed that the fermentation process determined many modifications of the physicochemical composition, the volatile and non-volatile profiles of the mango juices, so we can hypothesize that also the sensory characteristics would be very different. In this study it was not possible to carry out a sensory analysis, but the authors should consider these aspects in the future, realizing the essays in adequate quantities. Furthermore, if future experiments were carried out at an industrial or semi-industrial production level, the fermented mango juices could be subjected to consumer judgment, illustrating their anti-oxidant properties and investigating their predisposition to purchase and consume these juices.

The authors should discuss these two points in the introduction and in the conclusions with related references.

Response:

Thank you for your kind advice. We have supplemented these two points in the introduction and conclusions with related references in the revised manuscript. (Line 56-58, 60-62, and 453-458).

Comment 3:

Other comments:

2.3 Line 112: please, indicate the condition of sterilization of the mango juice used as control.

Response:

Thank you for your approval. We have supplemented the condition of sterilization of the mango juice in the revised manuscript. (Line 115)

Comment 4:

2.9 Statistical analyses

The authors did more statistical analyses than those described in this chapter: PCA hierarchical clustering analysis, PLS.

Please, integrate the information regarding the statistical analyses

Response:

Thank you for your approval. We have integrated the information (PCA, hierarchical clustering analysis, PLS-DA, OPLS-DA, SPLS-DA, volcano plot, and KEGG enrichment pathways) regarding the statistical analyses in the revised manuscript. (Line 224-229)

Comment 5:

Line 360: add the use of ESI in materials and methods.

Response:

Thank you for your kind advice. We have supplemented the detail inflammation of ESI in the revised manuscript. (Line 215-219)

Comment 6:

Line 379: better to add the meaning of VIP value.

Response:

Thank you for your kind advice. The full name of VIP is Variable Importance in Project. We have supplemented the full name of VIP in the revised manuscript. (Line 397-398)

Comment 7:

English language:

The English language is clear and fluid reading. There are only some minor revisions.

Response:

We sincerely thank the detailed comments from the reviewer. We have carefully corrected the grammatical mistakes throughout the revised especially for the proposed lines.

Comment 8:

Line 28: what do you mean by “upregulated” and “downregulated”?

Maybe, do you mean that their content increased (“upregulated”) and decreased (“downregulated”) ?

Verify the use of these terms in all the paper, please.

Response:

Thank you for your kind advice. We have corrected this erroneous statement in the revised manuscript. (Line 27, 401, 402, 405, 411, and 419)

Comment 9:

Revise the tenses of the English verbs in the paper, sometimes the authors use the present perfect, but it would be better using the past simple, for example, line 41 and line 58 have confirmed better confirmed Line 341: has also found better also found Line 442 have read better read

Response:

Thank you for your kind advice. We have revised the tenses of the English verbs in the revised manuscript (Line 39, 58, 68, 358, and 462-463).

Comment 10:

Minor revisions:

In the whole paper verify if there is a space before the citation number, in many cases there is not.

Response:

Thank you for your kind advice. We have added a space before the citation number in the revised manuscript.

Comment 11:

Line 58: sugars7. Correct the citation number and add a space between sugars and [7]: sugars [7].

Response:

Thank you for your kind advice. We have corrected the citation number in the revised manuscript. (Line 56)

Comment 12:

Line 413: verify [t39], maybe [39].

Response:

Thank you for your kind advice. We have corrected the citation number in the revised manuscript. (Line 431)

Round 2

Reviewer 3 Report

Comments and Suggestions for Authors

Dear authors,

thanks for your modifications.

I think it is possible to accept your paper for pubblication, after some minor revisions of the English language. (see the following session)

Comments on the Quality of English Language

Some  revisions of the English language are needed, here are the most relevant:

Lines 56-58: please revise this sentence.

The changes in the physicochemical composition, the volatile and non-volatile compounds of the mango juices with microbial fermentation, leading to the alteration in sensory characteristics [8].

Here is a suggestion:

The changes in the physicochemical composition, in the volatile and non-volatile compounds of the mango juices with microbial fermentation, also suggest a modification of the sensory characteristics [8].

Lines 60-62:  please revise this sentence.

Which may be an important reason that affects consumers' purchase and consume of juice at an industrial or semi-industrial production level 61 [11]. 

Change the sentence, here is a suggestion:

These can be important elements that affects consumers' purchase preferences and consume of fermented mango juice produced at an industrial or semi-industrial level [11]. 

 Line 68: been used to the fermentation of fruits and vegetables. 

This is not correct, maybe have been used for the fermentation.... 

Line 287-288: destroy....maybe destroy(third person singular)

 Lines 418-420: However, the sensory analysis of mango juice with/without L. rhamnosus FJG1530 fermentation weren’t implemented due to the independent of the number of samples.

Thise sentence is not clear, maybe you meant:

However, the sensory analysis of mango juice with/without L. rhamnosus FJG1530 fermentation could not be realized due to the limited size of the samples (100 mL).

Line 421: investigating their predisposition to purchase and consume L. rhamnosus FJG1530 fermented mango juice at an industrial or semi-industrial production level. 

change the sentence, here is a suggestion:

....juice, and to investigate consumers' predisposition to purchase and drink L. rhamnosus FJG1530 fermented mango juice, produced at industrial or semi-industrial level. 

Author Response

Comment 1:

Some revisions of the English language are needed, here are the most relevant:

Lines 56-58: please revise this sentence.

The changes in the physicochemical composition, the volatile and non-volatile compounds of the mango juices with microbial fermentation, leading to the alteration in sensory characteristics [8].

Here is a suggestion:

The changes in the physicochemical composition, in the volatile and non-volatile compounds of the mango juices with microbial fermentation, also suggest a modification of the sensory characteristics [8].”

Response:

Thank you for your kind advice. We have modified the sentence in this manuscript according to your comment. (Line 56-58)

Comment 2:

Lines 60-62:  please revise this sentence.

Which may be an important reason that affects consumers' purchase and consume of juice at an industrial or semi-industrial production level 61 [11].

Change the sentence, here is a suggestion:

These can be important elements that affects consumers' purchase preferences and consume of fermented mango juice produced at an industrial or semi-industrial level [11].

Response:

Thank you for your kind advice. We have modified the sentence in this manuscript according to your comment. (Line 60-62)

Comment 3:

Line 68: been used to the fermentation of fruits and vegetables.

This is not correct, maybe have been used for the fermentation....

Response:

Thank you for your kind advice. We have modified the sentence in this manuscript according to your comment. (Line 68)

Comment 4:

Line 287-288: destroy....maybe destroys (third person singular)

Response:

Thank you for your kind advice. We have modified the sentence in this manuscript according to your comment. (Line 289)

Comment 5:

Lines 418-420: However, the sensory analysis of mango juice with/without L. rhamnosus FJG1530 fermentation weren’t implemented due to the independent of the number of samples.

Thise sentence is not clear, maybe you meant:

However, the sensory analysis of mango juice with/without L. rhamnosus FJG1530 fermentation could not be realized due to the limited size of the samples (100 mL).

Response:

Thank you for your kind advice. We have modified the sentence in this manuscript according to your comment. (Line 420-422)

Comment 6:

Line 421: investigating their predisposition to purchase and consume L. rhamnosus FJG1530 fermented mango juice at an industrial or semi-industrial production level.

change the sentence, here is a suggestion:

....juice, and to investigate consumers' predisposition to purchase and drink L. rhamnosus FJG1530 fermented mango juice, produced at industrial or semi-industrial level.

Response:

Thank you for your kind advice. We have modified the sentence in this manuscript according to your comment. (Line 423-425)